# Surgical Technique of Central and Over-the-Top Full-Endoscopic Decompression of the Cervical Spine: A Technical Note

**DOI:** 10.3390/jpm13101508

**Published:** 2023-10-18

**Authors:** João Paulo Machado Bergamaschi, Marcelo Botelho Soares de Brito, Fernando Flores de Araújo, Ricardo Squiapati Graciano, Edgar Takao Utino, Kai-Uwe Lewandrowski, Fernanda Wirth

**Affiliations:** 1Atualli Spine Care Clinic, 745 Alameda Santos, Cj. 71, São Paulo 01419-001, Brazil; fernando_araujo87@clinicaatualli.com.br (F.F.d.A.); ricardosg.rsg@clinicaatualli.com.br (R.S.G.); 2Atualli Academy, 2504 Brigadeiro Luís Antônio, Cj. 172, São Paulo 01402-000, Brazil; edgar_utino@hotmail.com (E.T.U.); fernanda@atualliacademy.com (F.W.); 3Clínica Articulare, 791 Avenida Senador Lemos, Sala 301 a 305, Belém 66050-005, Brazil; mbrito@clinicaarticulare.com.br; 4Center for Advanced Spine Care of Southern Arizona, Tucson, AZ 85712, USA; business@tucsonspine.com

**Keywords:** over-the-top, central, full-endoscopic, decompression, cervical, spine, technical note

## Abstract

Endoscopic surgery of the cervical spine is constantly evolving and the spectrum of its indications has expanded in recent decades. Full-endoscopic techniques have standardized the procedures for posterior and anterior access. The full-endoscopic approach was developed to treat degenerative diseases with the least possible invasion and without causing instability of the cervical spine. The posterior full-endoscopic approach is indicated for the treatment of diseases of the lateral part of the vertebral segment, such as herniations and stenoses of the lateral recess and vertebral foramen. There has been little discussion of this approach to the treatment of central stenosis of the cervical spine. This technical note describes a step-by-step surgical technique for central and over-the-top full-endoscopic decompression in the cervical spine, using a 3.7 mm working channel endoscope. This technique has already been shown to be effective in a recent case series with a 4.7 mm working channel endoscope, and may represent a new treatment option for central or bilateral lateral recess stenosis. There is also the possibility of a bilateral full-endoscopic approach, but this may be associated with greater muscle damage and a longer operative time. Case series and comparative studies should be encouraged to confirm the safety and utility of this technique.

## 1. Introduction

Cervical radiculopathy due to foraminal pathology, either degenerative narrowing of the foramen or disc herniation, is a common condition [1,2]. The radicular symptoms in arm pain due to degenerative changes in this segment usually arise from osteophytes in the intervertebral foramen, facet hypertrophy, or from lateral disc herniations. In these cases, surgical decompression may be required if conservative treatment fails or if there is a neurological deficit [1]. Regarding this situation, full-endoscopic spine surgery (FESS) is considered an evolution of minimally invasive tubular spine surgery [3]. Originally, FESS was based on an endoscope with a continuous irrigation and working channel for lumbar transforaminal surgery [4], which evolved to other approaches and in other segments of the spine [5]. FESS has been shown to be effective in the lumbar spine [6,7,8,9,10], whereas there is a lack of data to support its role in cervical decompression, as most reports focus on discectomy and foraminal decompression [11,12]. Endoscopic surgery of the cervical spine is constantly evolving and the spectrum of its indications has expanded in recent decades. Various parameters such as rehabilitation time, bleeding, postoperative pain, length of hospital stay, and surgical incision, have shown superiority over conventional cervical spine surgery [13,14,15,16]. However, complications such as transient root paralysis or neuropraxia may be more common in this type of access. In the cervical spine, the application of minimally invasive techniques has been difficult because of the anatomical conditions, mainly in the intervertebral space and the foramen, in addition to the impossibility of moving the spinal cord [17]. The full-endoscopic techniques in cervical spine have standardized procedures for posterior and anterior approaches, and were developed to treat degenerative diseases with the least possible invasion, and without causing instability of the cervical spine [1,17,18].

Although surgical treatment via anterior discectomy and fusion (ACDF) is still a good possibility in the treatment of cervical disc herniations [2,19,20], this technique carries many risks such as pseudoarthrosis, infection, persistent pain, adjacent segment disease, and some complications associated with the approach such as dysphagia, vascular damage, and dysphonia, which sometimes cannot be avoided [2,21,22]. For this reason, some alternative procedures have been developed, such as microendoscopic, endoscopic-assisted, and full-endoscopic posterior foraminotomy [2,23]. The posterior procedures were developed with the aim of preserving the mobility and stability of the cervical segment. However, the main problem with this approach is the long learning curve [2]. Important indications for this procedure are disc herniations and bone stenoses in the lateral portion of the vertebral segment, such as in the lateral recess and foramen [24]. The more lateralized translaminar approach has been recommended for access to the most central herniations, which have traditionally been a major limitation for this approach [25]. Access to the posterior arch with the “full-endoscopic” technique is feasible but has not been discussed in the literature. The major concern when performing central endoscopic decompression of the cervical spine is the risk of spinal cord injury from manipulation of the working cannula, drill, and other instruments. This poorly performed maneuver can result in severe and irreversible neurologic damage [26].

Furthermore, the posterior approach may present other complications, such as bleeding, transient or persistent neurological injury, and axial neck pain [2,26]. Thus, there are mixed data suggesting that posterior approaches with disc manipulation may lead to the subsequent development of cervical kyphosis, especially in patients with preexisting loss of cervical lordosis [24,27], but Won et al. [28] have shown that posterior endoscopic foraminotomy was effective in reducing radicular symptoms and that there was no significant worsening of sagittal alignment in patients with a pre-existing loss of cervical lordosis [28].

Considering the development of posterior cervical endoscopic techniques, the aim of this study is to describe the surgical technique of central and over-the-top decompression.

## 2. Surgical Technique

We performed the full-endoscopic cervical procedure via a posterior approach with the patient in the prone position under general anesthesia and intraoperative neurophysiologic monitoring with mild flexion of the cervical spine. The puncture site was confirmed using fluoroscopy in the anteroposterior view (using the lateral line of the cervical interlaminar bone window) and a lateral view (parallel to the inclination of the disc to be reached). We performed a 7 mm cross-section with opening of the fascia and insertion of the dilator and working cannula, always watching for bone contact with the lateral edge of the bone window or the medial edge of the facet joint. The dilator and working cannula must be used as an instrument to detach the muscles of the superior and inferior lamina at the level to be treated. This step can also be performed with trephination through the working cannula. It is important to note that in a central or over-the-top decompression procedure, soft tissue cleaning must be performed to the medial border, i.e., at the base of the spinous processes. A working channel endoscope (ENDOLINE^®^, Ribeirão Preto, SP, Brazil) with a diameter of 3.7 mm was used with bipolar cautery and micropunch to remove soft tissues and visualize the juxtaposed edges of the laminae.

The first step of such a surgical approach must be the identification of point V or point Y (Figure 1), which represents the lateral edge of the interlaminar bone window. Decompression of the bone must be initiated from the superior lamina using a drill. We used the diameter of the drill itself (3.5 mm) to calculate the extent of bone resection of the inferior articular process of the cranial vertebra. We advocate for preservation of at least 50% of the facet joint to avoid possible postoperative iatrogenic instability. Although there are no in vivo studies demonstrating this relationship, some publications on cadavers show that resection of more than 50% of the facet joint can result in increased posterior deformity, decreased torsional stiffness, and segmental hypermobility [29,30].

The next step of bone decompression is the resection of the upper part of the lamina of the caudal vertebra, following the same rules as for hemilaminectomy of the cranial vertebra. So far, the steps are almost the same as in full-endoscopic posterior foraminotomy. In central decompression, from now on, we focus on the resection of the base of the caudal and cranial spinous processes, following the same principles as in full-endoscopic lumbar decompression [31,32]. In contralateral decompression, only the drill or kerrisson is inserted into the spinal canal (Figure 2). Bone resection at the base of the spinous process should be sufficient for instrument handling in the spinal canal without manipulating the spinal cord. The working cannula and endoscope remain outside the canal, floating and not touching or manipulating the spinal cord. All bone decompression can be performed without opening the ligamentum flavum, providing a protective mechanical barrier for the spinal cord. Before opening the ligamentum flavum, we recommend performing an ipsilateral foraminotomy if necessary. At this stage, pedicle-to-pedicle (caudal and cranial) decompression is the goal, and the most important structure to partially resect is the superior articular process (SAP) of the caudal vertebra, as it is the primary structure responsible for foraminal or lateral recess stenosis (Figure 3).

During foraminotomy, the ligamentum flavum is usually detached from the bone. After opening the entire ligamentum flavum, further bleeding is expected due to the presence of a venous plexus between this structure and the spinal cord, so prophylactic hemostasis with bipolar is a fundamental step to prevent further venous bleeding. After resection of the ligamentum flavum, a complementary contralateral bone resection may be required. The contralateral decompression parameters can be divided into anatomic, clinical, and radiologic parameters. The anatomic parameters include direct visualization of the contralateral root, presence of epidural fat pulsating in the contralateral lateral recess, absence of bony structures touching or compressing the medulla or contralateral root, and adequate contralateral space for use of a probe between the contralateral laminae and the spinal cord. Clinical parameters include improvement in intraoperative neurophysiologic monitoring after decompression, but depending on chronicity and damage before surgery, we may not see immediate improvement. Radiological parameters are confirmed by radioscopy and include decompression at the superomedial border of the contralateral caudal pedicle and decompression at the inferomedial border of the contralateral cranial pedicle, according to Table 1. Table 1 provides parameters for satisfactory contralateral decompression according to Bergamaschi et al. [33].

Posterior cervical over-the-top decompression, as described, allows for satisfactory decompression in postoperative examinations (Figure 4).

## 3. Discussion

In 1996, Spetzger et al. [34,35] proposed the first unilateral laminotomy for a bilateral approach to the spinal canal with the aim of decompressing the lumbar spine. Subsequently, over-the-top decompression was combined with tubular retractors. A major advantage of this technique is that it allows for preservation of the posterior osseoligamentous complex and, therefore, has a lower chance of iatrogenic instability when compared with conventional laminectomies [36].

The development of minimally invasive spine surgery has made it possible to perform decompression with the full-endoscopic technique, initially in the lumbar spine. Ruetten et al. [1] showed good results with a reduction in pain intensity and indices of functional disability using the Oswestry questionnaire. Other authors [2,6] also demonstrated good results with few complications. The main difficulty in performing the technique is the long learning curve and the surgical time, which can be very long in the first 20 cases. An experienced surgeon can perform the same decompression via a tubular or minimally invasive approach in less time. After reaching the plateau of the learning curve, the full-endoscopic technique can be performed with a surgical time comparable to that of other techniques but with less bleeding, a shorter hospital stay, less postoperative pain, and a shorter rehabilitation time [1,2,6]. After several years, some authors [37,38,39] demonstrated the effectiveness of the technique on the thoracic spine, with an improvement in pain intensity and neurological deficit in patients with hypertrophy or calcification of the ligamentum flavum. The presence of the spinal cord may at first glance cause concern among spine surgeons, but the full-endoscopic technique may be a good option in the treatment of posterior central compression in the thoracic spine, with few reported complications [37,38,39]. Similarly, we can apply this concept to the cervical spine, but there has been little discussion of central and over-the-top cervical decompression here; however, in the hands of experienced endoscopic surgeons, it may be a good option for the treatment of central stenosis.

Patients aged approximately 60 years or older with degenerative cervical spinal stenosis are at high risk for cervical spondylotic myelopathy. Cervical unilateral laminotomy for bilateral decompression (ULBD) has been studied [40] and has become a major surgical technique in minimally invasive surgery (MIS), resulting in an excellent reduction of the Oswestry Disability Index, and back and leg pain, while it is associated with a low complication rate [40,41,42]. More recently, it has been described in full-endoscopic decompression surgery using a 4.7 mm working channel endoscope [6]. The ULBD “over-the-top” technique can be used in patients with cervical spondylotic myelopathy without evidence of instability [40]. This full-endoscopic technique was used prospectively in 10 elderly patients in a case series. The patients in this cohort had unique and complex pathologies with cervical spinal stenosis, particularly due to compensatory hyperlordosis associated with thoracic kyphosis. All patients in the cohort had preoperative neck pain (visual analogue scale (VAS) average 5.8 ± 0.9), which was associated with impaired hand dexterity in most patients (9/10 patients). The authors followed the patients for an average of 22 ± 4.7 months. During this period, Nurick grades and the modified Japanese Orthopedic Score (mJOA) improved significantly compared with preoperative values. Patients also showed a trend toward improvement in their VAS neck pain score, which was 5.8 ± 0.9 during the preoperative period and dropped to 2.9 ± 0.6 at the last follow-up visit [6].

The evolution from posterior open cervical foraminotomy to tubular, microendoscopic, and FESS, is associated with advantages, such as less surgical muscle aggression, a smaller incision, shorter hospital stay, less postoperative pain, and a shorter rehabilitation time [43], as traditional open cervical laminectomy is correlated with the possibility of instability and delayed kyphosis [40,44,45,46]. Another advantage of full-endoscopic cervical spine surgery is the continuous saline irrigation, which can lead to the reduction of inflammatory agents and thus less intraoperative bleeding, resulting in a lower risk of dural tears due to the clear surgical field [47,48]. Although biomechanical and clinical studies comparing these techniques to traditional open techniques in terms of long-term outcomes have not been performed, reports in the literature from other segments of the spine suggest that the risk of infection after decompressions, length of hospital stay, risk of kyphosis increase, secondary need for fusion, and postoperative narcotic dependence is reduced [40].

Regarding indications, full-endoscopic cervical spine surgery may be indicated for many pathologies. Initially it was used for disc herniation and foraminal stenosis, but the development of the technique allows for its use in cases of cervical spondylotic myelopathy, including central canal stenosis, cervical disc herniation and calcified ligamentum flavum. Therefore, there is a significant improvement in all aspects of pain relief, imaging results, and functional outcomes, during midterm follow-up after full-endoscopic cervical spine surgery [48]. Anterior cervical discectomy and fusion has long been considered the gold standard for the treatment of cervical myelopathy or radiculopathy [48,49,50], however, the anterior approach can lead to serious complications such as injury to the esophagus, trachea, and neurovascular bundle during the procedure [48]. Full-endoscopic anterior cervical discectomy is indicated when a herniated disc is unresponsive to conservative treatment and there is an annular tear with consistent pain on provocative discography. [48,51,52]. In contrast, migrated disc herniation, a collapsed disc space < 5 mm, calcified disc, infection, instability, and previous anterior cervical surgery are contraindications. For this reason, posterior approaches for cervical decompression have generally been chosen for the treatment of multilevel cervical disc herniations and ossifications of the posterior longitudinal ligament [48,53]. In open spine surgery, the posterior cervical approach can lead to complications such as cervical instability, significant blood loss, persistent neck pain, and postoperative kyphosis in elderly patients because of extensive muscle dissection [48,54]. Full-endoscopic posterior cervical foraminotomy is indicated for foraminal disc herniations with largely single lateral arm pain, foraminal stenosis with unilateral arm pain (single or multilevel), and persistent symptoms in patients with previous anterior cervical discectomy. Contraindications, on the other hand, include instability and only axial neck pain [55,56].

Despite the advantages and indications of full-endoscopic techniques, the main limitation in the development of this technique is the long learning curve associated with less invasive techniques and the greater number of complications observed during this period [57]. Because of the minimal working channel, this technique has a limited operative field of view, so the spinal cord may be injured by the instruments, especially with inexperienced surgeons [48,58]. Sclafani et al. observed a 41% reduction in operative time after 10 posterior cervical full-endoscopic decompressions [57]. The learning curve for cervical endoscopy is shorter than the curve for lumbar endoscopy because generally only a surgeon experienced in lumbar endoscopic techniques will progress to cervical endoscopic techniques, and this shorter operative time reflects a lower incidence of complications. In the same study, the authors observed the occurrence of complications such as durotomy, neurological damage, and conversion to open surgery in the first 30 cases of the different minimally invasive or percutaneous techniques.

The full-endoscopic central decompression technique represents one of the final stages of the learning curve in the various segments of the spine and must be performed by surgeons experienced in endoscopic techniques [59]. Cervical endoscopy should be started when the surgeon is fully proficient in the use of instruments such as the drill, bipolar, Kerrisson, and endoscope, because cervical endoscopy does not allow major manipulation of neurologic structures, so access to the foramen, disc, or contralateral side may be difficult in many cases. With technical improvement, it is possible to perform resection of posterior disc osteophytes [60], translaminar approaches [61], and access to central cervical hernias via the posterior approach [25]. However, the technique of full-endoscopic central cervical decompression is still little discussed in the literature.

With regard to other segments of the spine, full-endoscopic lumbar decompression is well described in the literature and represents a further development of the lumbar interlaminar approach [10,41,62,63,64,65]. Komp et al. [63] cite the advantages of the lumbar technique as including the following: ease of operation for the surgeon due to excellent visualization, good illumination, and an increased field of view with the endoscope of 25 degrees. In addition, they point to the cost-effectiveness of the procedure because of its short operative time, rapid rehabilitation, low cost of postoperative care, and less anatomic trauma [63]. Recently, Siepe et al. [66] described full-endoscopic bilateral over-the-top decompression for lumbar central stenosis, which represents the next step in reducing surgical invasiveness compared with microsurgical decompression. In addition, this procedure is clinically feasible, can be performed safely, and is an efficient surgical technique. This technique displayed better results in obese patients because the operations can be performed almost regardless of the patient’s weight. The rate of infections and wound healing problems are also significantly lower. However, this technique can be considered complex and should only be performed by surgeons who are at an advanced stage of the full-endoscopic learning curve [66].

Other recent cohort studies examining the postoperative complications and clinical outcomes of patients diagnosed with spinal stenosis who underwent full-endoscopic lumbar decompression or MIS had longer operative times for endoscopic decompression but a shorter hospital stay and better clinical outcomes. For example, patients treated with the full-endoscopic technique showed better improvement in low back pain after one year than patients who underwent minimally invasive tubular surgery [10]. This improvement is expected to occur gradually in other segments of the spine. An et al. described over-the-top full-endoscopic thoracic decompression as a treatment option for thoracic myelopathy due to ossification of the ligamentum flavum [64]. After surgery, there was significant improvement in the JOA (Japanese Orthopedic Association) score and the ASIA (American Spinal Injury Association Impairment Scale) score for sensibility and movement [64].

In our study, we provided detailed technical information to contribute to the advancement of the full-endoscopic technique for over-the-top cervical decompression and to further reduce the learning curve of endoscopic spine surgeons. The use of an endoscope with a smaller working channel (3.7 mm) may provide an advantage over the previously described technique using an endoscope with a working channel of 4.7 mm [6], because finer instruments are used to access limited and reduced spaces.

The possibility of a bilateral full-endoscopic approach can be considered as an option for the technique we have described, but may be associated with greater muscle damage and longer operative times, especially if the surgeon’s side and the equipment must be changed when the side of access is changed. This technique may be beneficial for obese and elderly patients in whom open surgery requires greater exposure and is considered riskier. Studies comparing the two techniques need to be conducted to validate their benefits.

## 4. Conclusions

Cervical over-the-top full endoscopy is a feasible technique and may represent a new treatment option for central or bilateral lateral recess stenosis. This technique may be beneficial for obese and elderly patients because it requires less exposure, less muscle damage and blood loss, a shorter hospital stay, and a rapid return to daily activities. Although there are mixed data suggesting that cervical kyphosis may occur in patients with preexisting loss of cervical lordosis, there are also clinical data suggesting that cervical lordosis does not worsen in this patient population. In addition, complication rates are lower with posterior approaches than with anterior approaches. Case series, clinical trials, and comparative studies should be encouraged to confirm the safety and benefits of this technique.

## Figures and Tables

**Figure 1 jpm-13-01508-f001:**
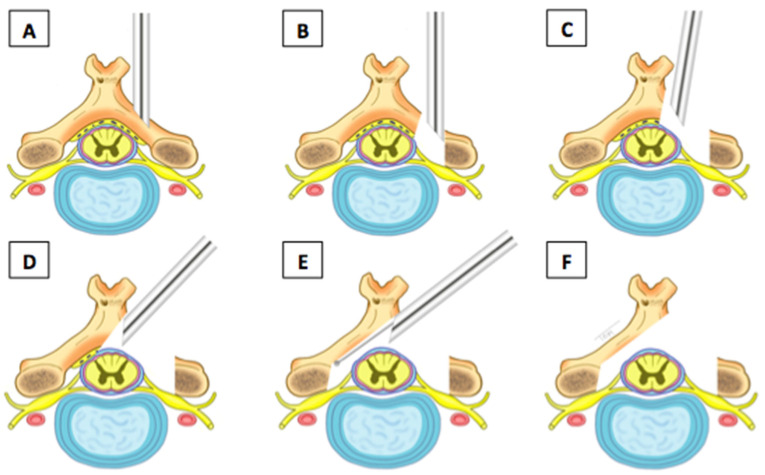
Step-by-step instructions for central and over-the-top percutaneous endoscopic decompression. Decompression of the superior and inferior lamina from the level of attachment (**A**–**C**) to the base of the spinous process (**D**). Bone resection and contralateral yellow ligament (**E**) for central decompression and contralateral lateral recess (**F**).

**Figure 2 jpm-13-01508-f002:**
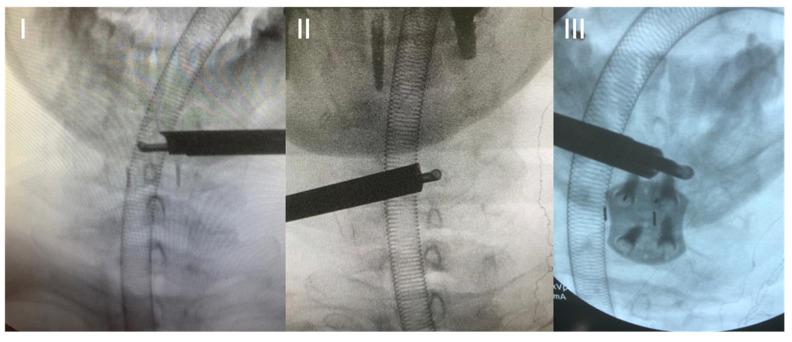
Intraoperative radioscopic images. Decompression of the base of the spinous process (**I**,**II**). Positioning of the working cannula and instrument during contralateral decompression (**III**).

**Figure 3 jpm-13-01508-f003:**
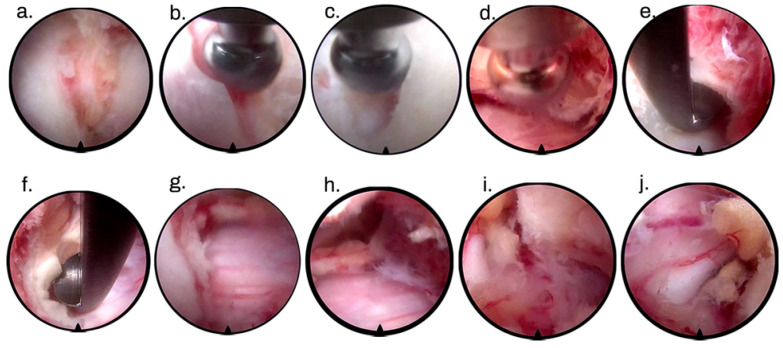
Intraoperative images of central and over-the-top cervical full-endoscopic decompression. Point identification of V or Y (**a**). Opening of the cranial (**b**) and caudal (**c**) lamina. Decompression of the base of the spinous process (**d**). Opening of the contralateral caudal (**e**) and cranial (**f**) lamina. Final aspect of contralateral decompression (**g**,**h**). Ipsilateral foraminotomy surrounding the caudal pedicle (**i**), and final image of the ipsilateral foraminotomy (**j**).

**Figure 4 jpm-13-01508-f004:**
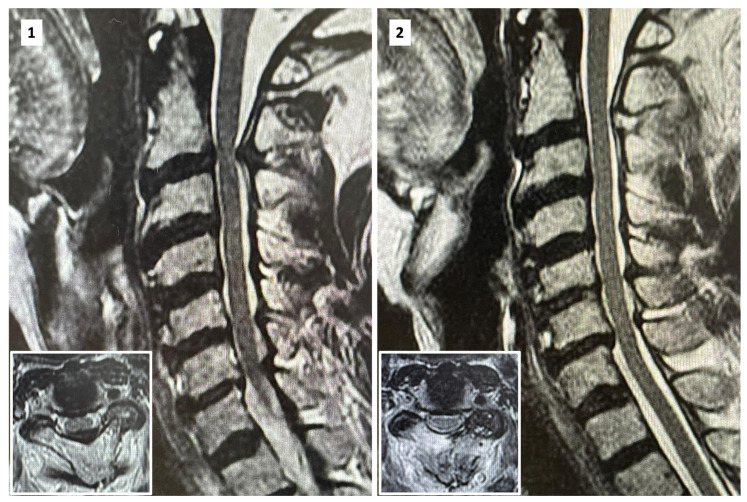
Preoperative (**1**) and postoperative (**2**) images of central and over-the-top cervical (C2-C3) full-endoscopic decompression.

**Table 1 jpm-13-01508-t001:** Parameters of satisfactory contralateral decompression, according to Bergamaschi et al., 2023 [33].

Radiographic Parameters Decompression at the superomedial border of the contralateral caudal pedicle. Decompression at the inferior-medial border of the contralateral cranial pedicle.
Clinical Parameters Improvement of somatosensory evoked potential.
Anatomical Parameters Direct visualization of the contralateral root. Presence of epidural fat pulsing in the contralateral lateral recess. Absence of bone structure in contact or compressing the medullar or contralateral root. Sufficient contralateral space for the use of a probe between the contralateral lamina and the spinal cord.

## Data Availability

The research data will be made available upon request by the corresponding author due to privacy.

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
