# Peer review of "Surgical Technique of Central and Over-the-Top Full-Endoscopic Decompression of the Cervical Spine: A Technical Note"

_jpm, 2023, doi:10.3390/jpm13101508_

Round 1

Reviewer 1 Report

This is a possible review and weakness of the article:

The article is a technical note that describes a novel surgical technique for cervical spine decompression using a full-endoscopic approach. The authors provide a detailed step-by-step description of the procedure, which involves resection of the base of the spinous processes and contralateral decompression of the spinal canal. They also suggest that this technique may have advantages over conventional or minimally invasive surgery, such as less muscle damage, shorter operative time, and better clinical outcomes.

1. The main weakness of the article is the lack of clinical evidence to support the authors’ claims.
2. The article is based on the authors’ experience and does not present any case series, comparative studies, or outcome data to demonstrate the safety and efficacy of the technique.
3. The article also does not  limitations, or contraindications of the technique.
4.  One should be careful in formulating conclusions. Rather, it would be necessary to evaluate the method's effectiveness and safety, and then present the technique with treatment results based on a series of cases.

Author Response

  1. The main weakness of the article is the lack of clinical evidence to support the authors’ claims.

Reply: There is case series published in 2020 by Carr et al. We cited this paper in our article.

  1. The article is based on the authors’ experience and does not present any case series, comparative studies, or outcome data to demonstrate the safety and efficacy of the technique.

Reply: There is case series published in 2020 by Carr et al. We cited this paper in our article.

  1. The article also does not  limitations, or contraindications of the technique.

Reply: We wrote about contraindications and limitations of the technique on lines 248- 253.

  1. One should be careful in formulating conclusions. Rather, it would be necessary to evaluate the method's effectiveness and safety, and then present the technique with treatment results based on a series of cases.

Reply: There is case series published in 2020 by Carr et al. We cited this paper in our article.

Reviewer 2 Report

The topic of the paper is in the general audience interest as the endoscopic approach became larger considered in the clinical practice. The paper is original as a technical note and represent an interesting report for development of further studies in the field. The paper is relatively well written and sentence are easily understandable.

The main observation concerns the introduction which must be more detailed and focused on the posterior approach. I suggest introduction will be addressed to history, indication and pathology of posterior approach only.

Figure 1 contains good explication of the surgical technic end useful to understand the philosophy of the approach.

Size and quality of pictures included in figure 3 is poor and if possible should be improved.

Discussion contain significative suggestions regarding the learning curve and the comparison with lumbar surgery is adequate. The possibility of bilateral access may be omitted or reduced in this technical note.

Minor revisions

Author Response

Size and quality of pictures included in figure 3 is poor and if possible should be improved.

Reply: The size and quality of Figure 3 were improved.

Discussion contain significative suggestions regarding the learning curve and the comparison with lumbar surgery is adequate. The possibility of bilateral access may be omitted or reduced in this technical note.

Reply: The discussion was complemented with more information about the learning curve. All modifications are highlighted in yellow.

Reviewer 3 Report

thank you for the opportunity to review this paper.

Endoscopic spine surgery is not so common and for sure not well known to the general orthopedic community outside of the spine surgeon community.

I enjoyed reading the technique description and i believe that the description together with the images is more than  sufficient.

I do have one important comment. Although its a technique article and not a review i feel that the background should rely on a more up to date papers. most of the litarature in the referance list is from 2018 and south. 

A quick search showed me many papers describing endoscopic cervical spine surgery from 2023. i would like to see the most uptodate knowledge published now.

this paper is the only one i found about over the top technique, published in 2018, i think you should consider citing it also:

Hussain I, Schmidt FA, Kirnaz S, Wipplinger C, Schwartz TH, Härtl R. MIS approaches in the cervical spine. J Spine Surg. 2019 Jun;5(Suppl 1):S74-S83. doi: 10.21037/jss.2019.04.21. PMID: 31380495; PMCID: PMC6626755.

In summary: this is an interesting paper, i think the background should be revised

Author Response

Endoscopic spine surgery is not so common and for sure not well known to the general orthopedic community outside of the spine surgeon community.

I enjoyed reading the technique description and i believe that the description together with the images is more than  sufficient.

I do have one important comment. Although its a technique article and not a review i feel that the background should rely on a more up to date papers. most of the litarature in the referance list is from 2018 and south. 

A quick search showed me many papers describing endoscopic cervical spine surgery from 2023. i would like to see the most uptodate knowledge published now.

this paper is the only one i found about over the top technique, published in 2018, i think you should consider citing it also:

Hussain I, Schmidt FA, Kirnaz S, Wipplinger C, Schwartz TH, Härtl R. MIS approaches in the cervical spine. J Spine Surg. 2019 Jun;5(Suppl 1):S74-S83. doi: 10.21037/jss.2019.04.21. PMID: 31380495; PMCID: PMC6626755.

In summary: this is an interesting paper, i think the background should be revised

Reply: The background was revised and complemented with more information. The present reference was cited. All the modifications are highlighted in yellow.

Round 2

Reviewer 1 Report

I accept in present form.

Reviewer 3 Report

Dear authors,

Thank you for addressing the comment of the original version.

The corrections were made according to the comments of the original review therefore my recommendation is to accept the paper